# PM$_{10}$ variation, composition, and source analysis in Tuscany (Italy) following the COVID-19 lockdown restrictions

Fabio Giardi[1,2], Silvia Nava[1,2], Giulia Calzolai[2], Giulia Pazzi[1,2], Massimo Chiari[2], Andrea Faggi[1], Bianca Patrizia Andreini[3], Chiara Collaveri[3], Elena Franchi[4], Guido Nincheri[4], Alessandra Amore[5], Silvia Becagli[5], Mirko Severi[5], Rita Traversi[5], and Franco Lucarelli[1,2]

[1]Department of Physics and Astronomy, University of Florence, Sesto Fiorentino (FI), 50019, Italy
[2]National Institute of Nuclear Physics (INFN), Sesto Fiorentino (FI), 50019, Italy
[3]Environmental Protection Agency of Tuscany - Regional Center for Quality Air Protection, Livorno, 57126, Italy
[4]Environmental Protection Agency of Tuscany - Central wide-area laboratory, Florence, 50144, Italy
[5]Department of Chemistry, University of Florence, Sesto Fiorentino (FI), 50019, Italy

**Correspondence:** Fabio Giardi (fabio.giardi@fi.infn.it)

**Abstract.** To control the spread of COVID-19, in March 2020 exceptional restrictive measures were taken imposing a radical change in the lifestyle of millions of citizens around the world, albeit for a short period. The national lockdown, which in Italy lasted from 10 March to 18 May 2020, was a unique opportunity to observe the variation in air quality in urban environments in a condition of almost total traffic block and a strong reduction in work activities. In this paper, the data from seventeen urban monitoring sites in Tuscany are presented by comparing PM and NO$_2$ of the two months before the start of the lockdown and the two after with the corresponding months of the previous three years. The results show that the total load of PM$_{2.5}$ and PM$_{10}$ decreased but it did not exhibit significant changes compared to previous years, while NO$_2$ undergoes a drastic reduction. For three of these sites, the chemical composition of the collected samples was measured by thermal-optical, ion chromatography, and particle-induced X-ray emission analysis and the application of multivariate Positive Matrix Factorization analysis allowed the PM$_{10}$ source identification and apportionment. Thanks to these analyses it was possible to explain the low sensitiveness of PM$_{10}$ to the lockdown effects as due to different, sometimes opposite, behaviors of the different sources that contribute to PM. The results clearly indicated a decline in pollution levels related to urban traffic and an increase in the concentration of sulfate for all sites during the lockdown period.

## 1 Introduction

Starting from the city of Wuhan, the capital of Hubei province of China, in early 2020, the SARS-CoV-2 virus quickly spread around the world becoming an established pandemic by the World Health Organization (WHO, 2021). The spread of the virus forced many countries around the world to impose severe restrictions on their population with temporary suspensions of travels, non-essential activities for citizens, and social aggregation in order to minimize physical contact between people and thus the spread of the acute respiratory disease (COVID-19) caused by the virus. A few weeks after its first confirmed case of SARS-CoV-2 in the country (confirmed on 30 January), Italy became one of the first countries to be heavily affected by the spread of COVID-19. Thus, the Italian government, following the example of the city of Wuhan, intervened in order to avoid the

overload of the national health system and to protect the health of the entire population, especially the most vulnerable people. Initially, after the rapid increase in cases of infection, admissions to intensive care and deaths in many provinces of the north of the country (mainly located in the Lombardy region) these were declared areas in isolation, from which it was not possible to

enter or exit and in which the countermeasures included stop to travels, closure of schools and non-essential businesses. Upon the worsening of the health situation also in the rest of the country, on 10 March 2020 the national lockdown was proclaimed (DPCM, 2021) and the restrictions of the most affected areas were extended to the whole country with the ban on gathering both outdoors and indoors. During the lockdown, which lasted until May 3, travels were only permitted for necessary activities, including health reasons, grocery shopping, and proven work needs. Therefore, this forced block of the population drastically

reduced numerous anthropogenic activities which, especially in inhabited centers, are a source of both gaseous and particulate air pollution.

Besides the desired result of containing the pandemic by lowering the infection curve (Signorelli et al., 2020), the restriction measures implemented through the lockdown may have brought other benefits, albeit temporary, to the health of the population in all the cities around the world that adopted the lockdown policy. In fact, the result was a period of almost total absence of

traffic both outside and inside the urban centers and the reduction of many industrial activities. This represents an extreme case hardly replicable in which at least one of the main sources of urban pollution is drastically reduced. This precious window allows us to study how the pollutants typically used to define air quality respond to a quick decline of anthropogenic emissions in terms of weeks' scale and offered also a valuable and unprecedented opportunity to assess how a significant abatement of road traffic impacted urban air quality.

Recently, many studies focused on the consequences of the lockdown on the air quality in many cities worldwide. Most of them concern large cities where vehicular traffic is one of the main sources of air pollution and their results involve the pollution indicators monitored by environmental regulatory agencies. In most cases ambient $PM_{10}$, $PM_{2.5}$, $NO_2$, $SO_2$, and CO concentrations decreased significantly during the COVID-19 lockdown either compared against the pre-lockdown period (Collivignarelli et al., 2020; Selvam et al., 2020; Chu et al., 2021; Connerton et al., 2020) or against the same period of

the previous years (Sharma et al., 2020; Jain and Sharma, 2020; Kerimray et al., 2020; Gualtieri et al., 2020). In all these scenarios, ozone behaved oppositely, showing a slight increase although cases have been reported in which its increase was much more pronounced (Lian et al., 2020; Hasim et al., 2021). These studies document that there have been improvements on a metropolitan or regional scale but there have been cases in which opposite situations occurred, such as the increase of some of these pollutants (Mor et al., 2021; Broomandi et al., 2020) or the non-decrease of ozone (Singh et al., 2020).

The large variation in these short-term findings can be ascribed to factors not related to COVID-19, such as meteorology and fluctuations in other regional emissions. In particular, meteorology may play an important role in the observed changes significantly influencing the atmospheric concentrations of the substances present in the atmosphere (Zhao et al., 2020a). In the northern hemisphere, for example, starting from March, with the arrival of spring, the air quality improves, mainly due to the enhanced dispersion capabilities of the atmosphere (e.g., the greater height of the mixing layer), except for the pollutants

produced by photochemical reaction, like sulfates and ozone, which increases with the warm months (Gerasopoulos et al., 2006). This indicates that the generation of secondary pollutants is not affected only by emission reduction but is influenced

by multiple factors that make the many studied cases hardly comparable. For this reason, the impact of traffic reduction on air quality is highly variable among different cities depending on the meteorological conditions (Xiang et al., 2020; Wang et al., 2020).

Furthermore, the increase in ozone highlights the essential role that secondary reactions play in the formation of fine particles. This is especially true in large urban centers since human health is more damaged by the presence of fine and secondary particles which have been estimated to be responsible for over eight million premature deaths per year (Burnett et al., 2018). The observed increase in ozone is closely related to the change in emissions that occurred in the cities. Nitrogen oxides (namely NO and $NO_2$), linked to traffic as a by-product of combustion, because of their short lifetime in the atmosphere decrease until

they are no longer able to neutralize the formation of ozone which, without the main removal mechanism and a comparable reduction of emissions of volatile organic compounds, start to accumulate in the atmosphere. This effect, known as the weekend effect (Sicard et al., 2020a; Schipa et al., 2009; Blanchard et al., 2008), is observable in large cities during weekends due to the drastic drop in vehicular traffic compared to weekdays but, in the COVID period, the increase in ozone levels was much more pronounced than the weekend effect (Sicard et al., 2020b). The consequence was an increased oxidative capacity of the

atmosphere due to the large presence of ozone which can lead to the formation of a large amount of secondary $PM_{2.5}$ and therefore to greater health risks.

Since many studies of air quality changes during the pandemic are based on networks of regulatory monitors, analyses of the chemical composition of particulate matter in the same period are still missing or sparse because they are not mandatory (Hicks et al., 2021; Wang et al., 2021a). Just a few cases estimated that variations occurred in the composition of the PM thanks

to the source apportionment obtained from receptor models (Dai et al., 2020; Zheng et al., 2020; Wang et al., 2021b).

Therefore, in this work, the comparison between 2020 and previous years of the main monitored pollutants was combined with the chemical analysis of the particulate matter to understand what changes occurred in air quality during the COVID-19 lockdown period and which have been the temporal variations of the main local sources in a metropolitan area where traffic is a predominant source of local air pollution. The comparison between a period of regular anthropogenic emissions and one

in which some of these have been highly reduced or almost completely removed can help to better distinguish the sources that contribute most to air pollution in cities and to investigate the impact that changes in primary pollutants emissions have on secondary chemical reactions. Such analyses can also be useful in view of possible future policy interventions aimed at reducing the load of particulate matter in urban centers.

## 2   Methods

### 2.1   Sampling

$PM_{10}$ daily mass concentrations before and after the national lockdown were studied in three areas and one settlement in Tuscany between January 1 and 30 April 2020. Several sampling sites were selected in each area for a total of seventeen air quality monitoring stations in the Tuscany air quality monitoring network managed by the Environmental Protection Agency of Tuscany. Sites and areas are shown in Table 1 and Fig. 1.

**Table 1.** List of the seventeen sampling sites.

| Area | Monitoring station | Abbrev. | Designation |
| --- | --- | --- | --- |
| Plain of Lucca | LU - Capannori | LU-1 | Urban background |
| | LU - San Concordio | LU-2 | Urban background |
| | LU - Micheletto | LU-3 | Urban traffic |
| Prato and Pistoia | PO - Ferrucci | PO-1 | Urban traffic |
| | PO - Roma | PO-2 | Urban background |
| | PT - Montale | PT-1 | Urban background |
| | PT - Signorelli | PT-2 | Urban background |
| Coast | LI - Cappiello | LI-1 | Urban background |
| | LI - Carducci | LI-2 | Urban traffic |
| | LI - La Pira | LI-3 | Urban background |
| | LI - ENI-Stagno | LI-4 | Urban background |
| Florentine agglomeration | FI - Bassi | FI-1 | Urban background |
| | FI - Boboli | FI-2 | Urban background |
| | FI - Gramsci | FI-3 | Urban traffic |
| | FI - Mosse | FI-4 | Urban traffic |
| | FI - Scandicci | FI-5 | Urban background |
| | FI - Signa | FI-6 | Urban background |

**Table 2.** Sampling periods of the subset samples used for $PM_{10}$ chemical speciation.

| Site | Period | Dates | Samples |
| --- | --- | --- | --- |
| FI-3 | BL | 24 February - 08 March 2020 | 14 |
| | DL | 16 March - 12 April 2020 | 28 |
| FI-1 | BL | 10 - 23 February 2020 | 14 |
| | DL | 16 March - 12 April 2020 | 23 |
| PO-2 | BL | 12 - 23 February 2020 | 12 |
| | DL | 16 - 22 March 2020, 06 - 19 April 2020 | 21 |

$PM_{10}$ daily mass concentrations were obtained in each sampling site by mass analyzers (FAI SWAM-DC; FAI SWAM-5A MONITOR) applying the method UNI EN 16450, automated continuous measurement systems (AMS) based on the use of $\beta$-ray attenuation whereas $PM_{2.5}$ concentrations were measured for eight of them with the same method.

    In three sites of the previous seventeen sampling sites (namely FI-3, FI-1, and PO-2), aerosol samples were collected on quartz fiber filters applying the method UNI 12341 on a daily basis (from midnight to midnight) by low volume ($2.3 \mathrm{~m^3~h^{-1}}$)
samplers (FAI HYDRA Dual Sampler), then the same filters were analyzed to obtain the chemical composition.

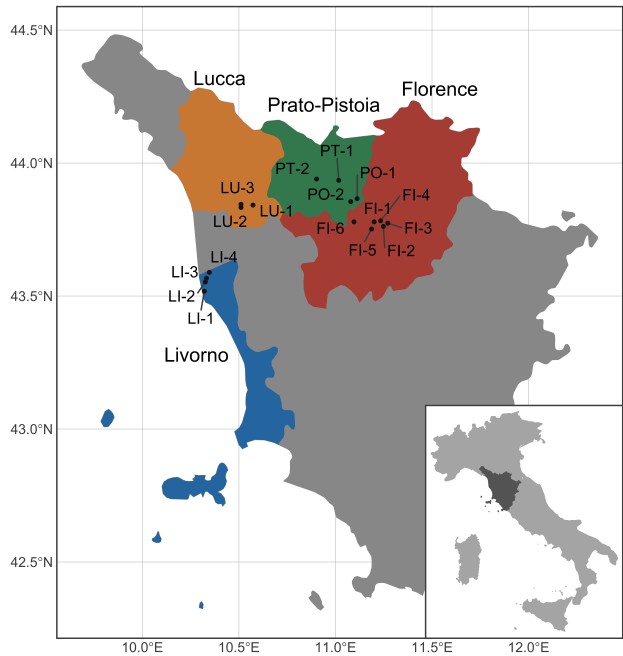

**Figure 1.** Position of the 17 sampling sites divided into the four metropolitan areas.

A total of 112 samples was collected in the period before the national lockdown (BL, i.e., before 9 March 2020) and in the period during the lockdown (DL). The observation periods are shown in Table 2 and, from these periods, the days 28 and 29 March are always excluded due to a very intense event of transport of dust from the Caspian Sea, which would lead to incorrect interpretations of the seasonal averages and source identification because it is not representative of the typical values of that period.

## 2.2 Chemical analysis

In order to perform two different chemical analysis techniques, the filters were cut in three parts. One was analyzed by ion chromatography (IC, Metrohm 930 Compact IC Flex) after the extraction in ultra-pure MilliQ water in a ultrasonic bath of the soluble component to quantify inorganic cations ($Na^+$, $K^+$, $Mg^{2+}$, $Ca^{2+}$, $NH_4^+$) and inorganic anions ($Cl^-$, $NO_3^-$, $SO_4^{2-}$) and applying the method UNI EN 16913. The quantification limits considered were calculated from the study of blank filters and the results are: $Cl^-$ 0.55 $\mu g\,m^{-3}$, $NO_3^-$ 0.157 $\mu g\,m^{-3}$, $SO_4^{2-}$ 0.181 $\mu g\,m^{-3}$, $NH_4^+$ 0.025 $\mu g\,m^{-3}$, $Na^+$ 3.8 $\mu g\,m^{-3}$, $K^+$ 0.091 $\mu g\,m^{-3}$, $Mg^{2+}$ 0.32 $\mu g\,m^{-3}$, $Ca^{2+}$ 0.74 $\mu g\,m^{-3}$.

The second part of the filter was analyzed by particle-induced X-ray emission (PIXE) which allowed the measurement of the concentrations of all the elements with atomic number Z > 16. The analyses were carried out with the 3 MV Tandetron accelerator of the LABEC laboratory in Florence with the external set-up described in Lucarelli et al. (2018, 2014), Chiari et al.

(2021), Calzolai et al. (2006). Using suitable membranes, like Teflon or Nuclepore filters, and advanced set-ups, this technique allows the analysis of elements with Z > 10; however, since for this work the particulate matter was collected on quartz filters, the high Si content of these substrata prevented the detection of elements from Na to P. Each sample was irradiated with a 3.0 MeV proton beam (10-150 nA intensity) for 90 s. The beam was collimated to about $2\,\text{mm}^2$ and filter scanning was carried out to analyze most of the deposit area. Elemental concentrations were obtained by a calibration curve from a set of thin standards. Measurement accuracy was tested using a NIST RM 8785 (National Institute of Standards and Technology, USA) standard. The minimum detection limits (MDLs) were of the order of few $\text{ng}\,\text{m}^{-3}$ for the low Z elements, down to a minimum value of 0.2 $\text{ng}\,\text{m}^{-3}$ for Cu-Zn. The total uncertainties on elemental concentrations were determined by the sum of independent uncertainties on certified thicknesses of the standards (5%), deposition area (2%), airflow (2%), and X-ray counting statistics (2%-20%). The uncertainties increase when concentrations approach minimum detectable limits (MDLs).

The third part of the filter was used to analyze the carbonaceous components of the particulate matter. Total, organic, and elemental carbon (TC, OC, EC) were analyzed by thermal-optical transmittance (TOT) analysis, using a Sunset Lab analyzer implementing a NIOSH-like protocol (reference method CEN/TC 264; Giannoni et al., 2016). Detection limits were 200 $\text{ng}\,\text{m}^{-3}$ while uncertainties were 5-10% for OC-TC and 10-20% for EC.

## 2.3 Receptor model

The concentration data obtained by the chemical analysis were used to perform a source apportionment study using the Positive Matrix Factorization (PMF) method. Briefly, PMF is a multivariate receptor modeling technique based on a weighted least square fit approach (Paatero and Tapper, 1994) which uses non-negativity constraints and weighs data values with realistic error estimates. The model uses the Eq. 1:

$$\mathbb{X}_{n\times m} = \mathbb{G}_{n\times p} \cdot \mathbb{F}_{p\times m} + \mathbb{E}_{n\times m} \tag{1}$$

where $n$ is the number of samples, $m$ the measured chemical species, $p$ the factors, $\mathbb{X}$ the matrix of the known concentrations of the species, $\mathbb{G}$ and $\mathbb{F}$ are matrices to be determined and which represent the contributions and the composition of the factors respectively, and $\mathbb{E}$ is the residual matrix, which is the difference between the real concentrations and those reconstructed by the model. The preparation of the input data and their uncertainties followed the guidelines suggested by Polissar et al. (1998). The model was applied to the whole dataset of the three sampling sites combined to increase the statistical significance of the analysis. The final analysis was carried out using the EPA PMF 5 software. Results for a varying number of factors were examined and, in order to find the most realistic solution, many parameters were observed, such as Q values (the values of the function minimized by the model), residuals distribution, the physical meaning of the factors (looking at the chemical profile and contributions), G-space plots, bootstrap, and displacement analysis.

Data analysis and visualization were carried out with R language (R Core Team, 2021).

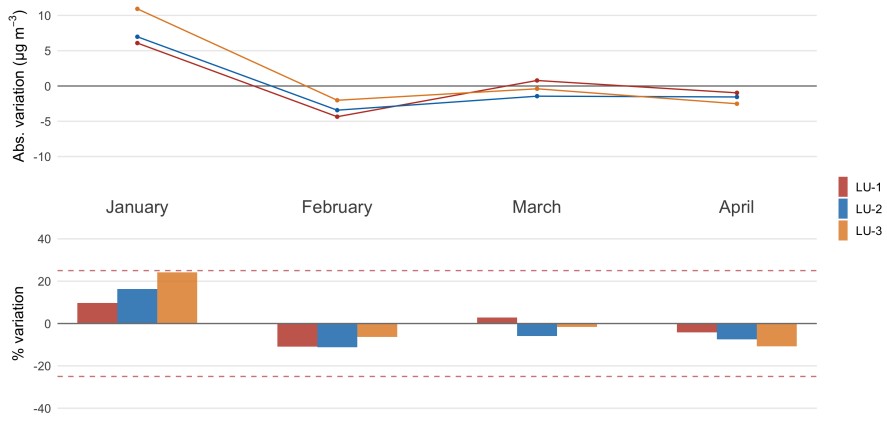

**Figure 2.** Plain of Lucca area. PM$_{10}$ absolute variation and percentage variation between monthly means 2020 and 2017-2019.

## 3 Results

### 3.1 Changes in PM concentration

The PM$_{10}$ concentration values in the stations covered by this study were evaluated with respect to the reference territorial context in the surroundings, three homogeneous areas of Tuscany (Plain of Lucca, Prato and Pistoia, Coast) and an agglomeration
(Florence). The data of each station were compared with the data of the stations present in the same homogeneous area.

The concentrations of PM$_{10}$ in March and April 2020, when the COVID-19 restrictions were applied, were assessed against the monthly mean values of the previous years, 2017-2019. The mean value of the three-year period 2017-2019 was taken as a reference. To verify the stationarity of the means with respect to this period, the months of January and February, not subject to the restrictions, are also taken into consideration.

Figures 2 to 5 show both the monthly variation percentages and the absolute differences in $\mu g\,m^{-3}$ for PM$_{10}$, detected in the stations of each area or agglomeration. A variation is considered significant if it exceeds 25%, so the range from -25% to + 25% is delimited by red dotted lines that refer only to percentage values.

### 3.1.1 LU stations

In the homogeneous Plain of Lucca area, PM$_{10}$ monitoring is carried out in the stations of LU-1, LU-2 (both urban background)
and LU-3 (urban traffic). In this area the PM values are homogeneous and high in the winter season. The annual means are lower than the limit set by the regulations ($40\,\mu g\,m^{-3}$) while the annual number of exceedances of the daily mean is larger than the limit value of 35 exceedances of the daily limit value, especially in the LU-1 station. Figure 2 shows a general consistency of the monthly means of PM$_{10}$ with respect to the value of the three-year period with the greatest positive changes in January.

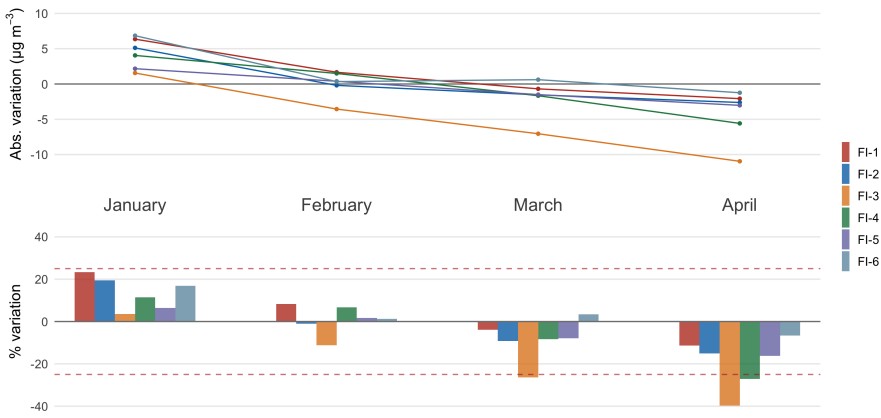

**Figure 3.** Florentine agglomeration. $PM_{10}$ absolute variation and percentage variation between monthly means 2020 and 2017-2019.

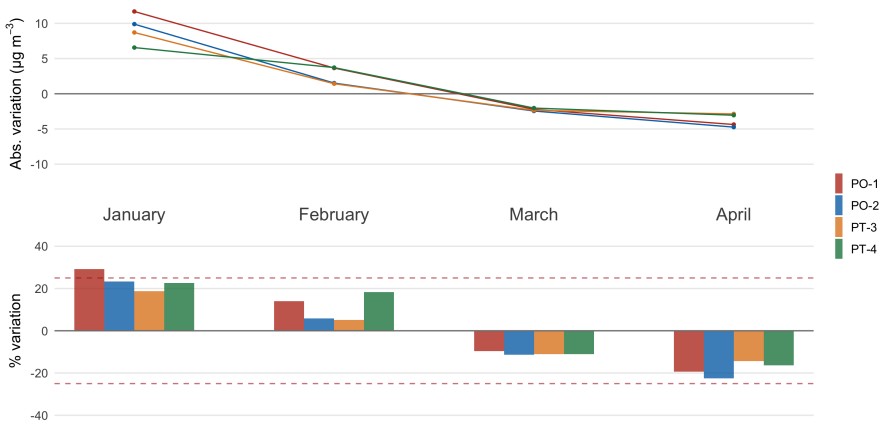

**Figure 4.** Prato-Pistoia area. $PM_{10}$ absolute variation and percentage variation between monthly means 2020 and 2017-2019.

In the other months, the monthly means vary with respect to the three-year period by a few units of $\mu g\,m^{-3}$ (Table 3) and the percentage variations are less than 25%.

### 3.1.2 FI stations

There are six $PM_{10}$ measurement stations in the agglomeration of Florence; two urban traffic (FI-4, FI-3) and four urban backgrounds (FI-1, FI-2, FI-5, FI-6).

The values recorded in January 2020 (Table 3 and Fig. 3) are higher than the previous three-year mean; the percentage variations, except for FI-1 (+26%), are however contained within 25%, even if they have a positive sign in all the stations, and the maximum absolute variation is around $6\,\mu g\,m^{-3}$ for the stations of FI-6 and FI-1. In February, the monthly means of 2020

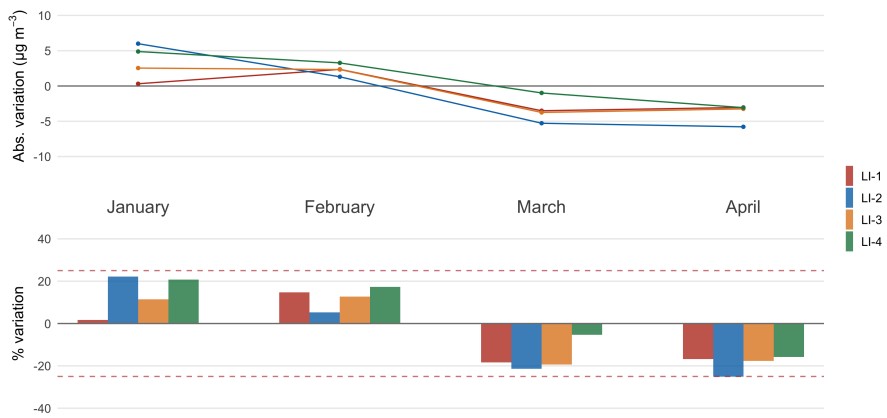

**Figure 5.** Livorno and coastal area. PM$_{10}$ absolute variation and percentage variation between monthly means 2020 and 2017-2019.

**Table 3.** PM$_{10}$ monthly mean concentrations ($\mu g\,m^{-3}$) for 2017-2019 and 2020 and percentage variation ($\Delta\%$).

| Site | January | | | February | | | March | | | April | | |
|------|---------|------|------|----------|------|------|-------|------|------|-------|------|------|
| | 2017-19 | 2020 | $\Delta\%$ | 2017-19 | 2020 | $\Delta\%$ | 2017-19 | 2020 | $\Delta\%$ | 2017-19 | 2020 | $\Delta\%$ |
| LU-1 | 62 | 68 | +10% | 40 | 35 | -13% | 28 | 29 | +4% | 23 | 22 | -4% |
| LU-2 | 43 | 50 | +16% | 31 | 27 | -13% | 24 | 23 | -4% | 21 | 19 | -10% |
| LU-3 | 45 | 56 | +24% | 32 | 30 | -6% | 25 | 25 | 0% | 23 | 21 | -9% |
| FI-1 | 27 | 34 | +26% | 20 | 22 | +10% | 17 | 17 | 0% | 18 | 16 | -11% |
| FI-2 | 26 | 31 | +19% | 19 | 19 | 0% | 17 | 15 | -12% | 17 | 15 | -12% |
| FI-3 | 43 | 45 | +5% | 32 | 28 | -13% | 27 | 20 | -26% | 28 | 17 | -39% |
| FI-4 | 35 | 39 | +11% | 22 | 24 | +9% | 20 | 18 | -10% | 21 | 15 | -29% |
| FI-5 | 34 | 36 | +6% | 22 | 23 | +5% | 19 | 17 | -11% | 19 | 16 | -16% |
| FI-6 | 41 | 48 | +17% | 26 | 26 | 0% | 18 | 19 | +6% | 19 | 17 | -11% |
| PO-1 | 40 | 52 | +30% | 26 | 30 | +15% | 23 | 21 | -9% | 22 | 18 | -18% |
| PO-2 | 42 | 52 | +24% | 26 | 28 | +8% | 22 | 19 | -14% | 21 | 16 | -24% |
| PT-1 | 46 | 55 | +20% | 28 | 29 | +4% | 21 | 19 | -10% | 20 | 17 | -15% |
| PT-2 | 29 | 36 | +24% | 20 | 24 | +20% | 18 | 16 | -11% | 19 | 16 | -16% |
| LI-1 | 19 | 20 | +5% | 16 | 18 | +13% | 19 | 16 | -16% | 18 | 15 | -17% |
| LI-2 | 27 | 33 | +22% | 25 | 26 | +4% | 25 | 19 | -24% | 23 | 17 | -26% |
| LI-3 | 22 | 25 | +14% | 18 | 21 | +17% | 19 | 16 | -16% | 18 | 15 | -17% |
| LI-4 | 24 | 28 | +17% | 19 | 22 | +16% | 19 | 18 | -5% | 20 | 16 | -20% |

are consistent with the previous three-year period and the most significant variation is that of FI-3 which has a mean lower than that of the three-year period of just over 3 $\mu$g m$^{-3}$, in percentage -11%. In March and April, the decrease in the FI-3 station is greater than 25%. Monthly means decrease from 27 to 20 $\mu$g m$^{-3}$ in March and from 28 to 17 $\mu$g m$^{-3}$ in April. The March and April values of the FI-3 station are similar to the values in the background stations for the same period. The other stations did not show significant changes in March, while in April a generalized decrease in values can be observed; however, the variations over 25% for this month were observed only in the FI-4 and FI-3 urban traffic stations.

### 3.1.3   PO stations

In the Prato-Pistoia area there are four reference stations for the measurement of PM$_{10}$: two in Prato (urban background PO-2 and urban traffic PO-1) and two in Pistoia, urban and suburban background. In the Prato-Pistoia area the variations in the monthly means from January to April have a very clear and consistent trend among all the stations. January and, to a lesser extent, February are characterized by monthly means higher than those of the previous three-year period, while March and April by lower monthly means. The variations are generally contained within 25% (with the sole exception of PO-1 in January) and the most significant reduction is observed in the PO-2 station in April which varies from a mean of 21 $\mu$g m$^{-3}$ in 2017-19 to a mean of 16 $\mu$g m$^{-3}$ in 2020 (Table 3, Fig. 4).

### 3.1.4   LI stations

In the coastal area of Livorno there are four reference stations (LI-1 to LI-4). In the Livorno area there are respectively limited increases in the monthly mean of PM$_{10}$ compared to the previous three years in January and February and limited reductions in March and April. The only station that at least partially reflects the decline in anthropogenic activities in the months of March and April is the LI-2 urban traffic station, which has shown a reduction close to 25% in the two months. The monthly means of PM$_{10}$ of the LI-2 station in 2020 are similar to those of LI-1 and LI-3, that is the urban background stations of the city, in the three-year period 2017-2019 (Table 3 and Fig. 5).

### 3.1.5   PM$_{2.5}$ throughout the regional network

Although we have a lot of data about PM$_{10}$ since it is both measured in all monitoring sites and collected on filters that can then be analyzed, for PM$_{2.5}$ we only have the mass for a small number of sites. Table 4 shows the monthly means in the three-year period 2017-2019 and in 2020 for the PM$_{2.5}$ monitoring stations in the study areas. A generally positive variation in January and a substantial stability of the means in the other months are observed. The month that records the greatest reductions is February 2020 with a -8 $\mu$g m$^{-3}$ in LU-1. In March and April, the variations are contained for all stations, always within $\pm$3 $\mu$g m$^{-3}$ regardless of the type of station.

Among the three stations whose PM$_{10}$ composition has been analyzed (FI-1 and FI-3 in Fig. 3, PO-2 in Fig. 4), those showing a significant variation of PM$_{10}$ in the months of March and April are the FI-3 station and to a much lesser extent the PO-2 station which, despite being an urban background, has always shown a good correlation with the traffic station located in

**Table 4.** PM$_{2.5}$ monthly mean concentrations (µg m$^{-3}$) for 2017-2019 and 2020 and percentage variation (Δ%).

| Site | January | | | February | | | March | | | April | | |
|------|---------|------|------|----------|------|------|-------|------|------|-------|------|------|
| | 2017-19 | 2020 | Δ% | 2017-19 | 2020 | Δ% | 2017-19 | 2020 | Δ% | 2017-19 | 2020 | Δ% |
| FI-1 | 20 | 28 | +40% | 14 | 14 | 0% | 11 | 12 | +9% | 10 | 10 | 0% |
| FI-3 | 26 | 30 | +15% | 19 | 16 | -16% | 14 | 13 | -7% | 14 | 12 | -14% |
| LI-1 | 11 | 13 | +18% | 10 | 67 | -30% | 9 | 9 | 0% | 9 | 10 | +11% |
| LI-2 | 17 | 22 | +29% | 14 | 11 | -21% | 13 | 10 | -23% | 12 | 11 | -8% |
| LU-1 | 55 | 56 | +2% | 33 | 25 | -24% | 20 | 22 | +10% | 15 | 16 | +7% |
| PO-1 | 32 | 40 | +25% | 19 | 16 | -16% | 14 | 13 | -7% | 13 | 12 | -8% |
| PO-2 | 34 | 42 | +24% | 20 | 17 | -15% | 14 | 13 | -7% | 12 | 11 | -8% |
| PT-1 | 40 | 46 | +15% | 23 | 21 | -9% | 15 | 14 | -7% | 13 | 12 | -8% |

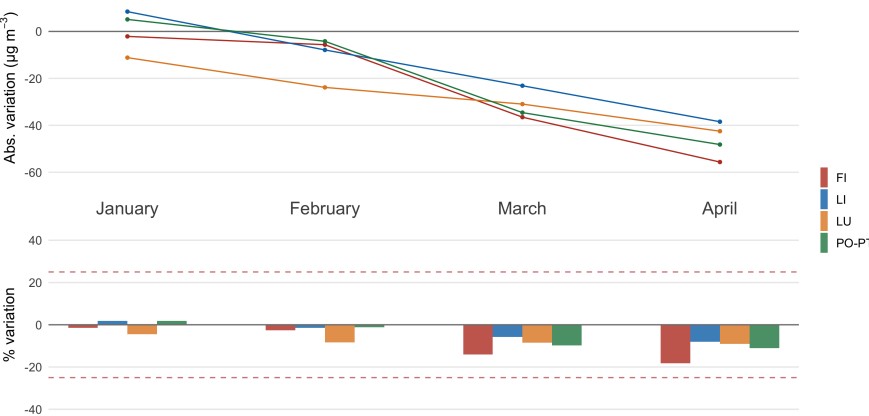

**Figure 6.** NO$_2$ absolute variation and percentage variation between monthly means 2020 and 2017-2019 in the four considered areas.

the same city. This correlation suggests that traffic is an important component of the background in that area. The only station with significant variations of more than 25% is FI-3 which can be considered the most characteristic as regards the traffic source at the regional level. As mentioned before, the FI-3 station in the months of March and April 2020 moves to the urban background values; moreover, interestingly there are no significant variations of PM$_{2.5}$ in this station, probably because the traffic decrease leads to a lower resuspension of urban dust which only affects PM$_{10}$.

## 3.2 NO$_2$ variation

In sixteen of the seventeen monitoring stations (except for FI-2) the measurement of the atmospheric concentration of nitrogen dioxide is also active and, also in this case, it was possible to compare the values of the first months of 2020 with those collected

in the three previous years. In all stations there is a net reduction of $NO_2$ concentration in the months of March and April. Figure 6 shows the comparisons for the monitoring stations grouped by metropolitan area. The greatest reduction during the months of lockdown is observed for the metropolitan area of Florence where the average decline is around -18% but with peaks of -40% and -22% respectively for the FI-3 and FI-4 stations, the most exposed to urban traffic. These results are in line with the $NO_2$ reduction observed in other regions and large cities affected by heavy traffic emissions (Baldasano, 2020; Keller et al., 2021; Huang and Sun, 2020; EEA, 2021). On the contrary, in the area of Lucca, despite the concentrations in 2020 are lower than in previous years, there is a much smaller difference between the first two months before the lockdown and the following ones. This is likely because this area has suffered less from the collapse of traffic as it is more industrial and less densely populated than Florence where traffic is generally more intense and some of the monitoring stations are located on large city avenues.

## 3.3 Changes in chemical composition

From the samples of the three sites dedicated to chemical analysis (FI-3, FI-1, and PO-2), combining the results of ion chromatography, PIXE and thermal-optical analysis, a dataset was obtained with a total of 33 chemical species, including the mass of $PM_{10}$. The analysis of such a large number of species, including the main and the trace, has the purpose of representing as much as possible the atmospheric particulate load and minimizing the unexplained fraction, as well as detecting the highest number of source markers. For each of the three sites, the samples were divided into two groups, as seen in Table 2. Taking as reference the 10 March 2020, the beginning of the national lockdown, the dates were split into two periods: before the lockdown and during the lockdown, respectively abbreviated to BL and DL. In fact, on the contrary of what we have seen so far regarding the variation of $PM_{10}$ and $PM_{2.5}$ concentrations in 2020 compared to the three years before the lockdown, as regards chemical analyses it was possible to make a comparison only within 2020. In fact, given the exceptional nature of the event, the chemical analyses were performed exceptionally on filters normally used only for environmental monitoring required by law that does not include routinely chemical speciation.

Table 5 shows the average concentrations in the atmosphere and the percentage variations between the two periods, whereas Fig. 7 shows the comparisons between the two periods of the value distributions through boxplots. In both cases, the major ones and the markers of the main urban sources are reported for each of the three sites.

As regards the total mass of particulate matter, on average in the observed periods all three sites are in line with the same slight reduction (from -25% to -28%). A similar decrease in the averages of the periods was observed for organic carbon (between 24% and 30%), while a stronger decrease was observed for the elemental fraction (between 37% and 66%), especially in the traffic site. Among the main species, a significant reduction is observed for Iron (between 52% and 81%) and, observing their distribution (Fig. 7), a very marked difference between the two periods is evident. In fact, in each site the concentration values of the period before the beginning of the lockdown are rarely exceeded by those of the following period. The same behavior is observed for some elements present at lower concentrations. In fact, Chromium and Copper have a spatial distribution similar to Iron and their decrease in the three sites is also close to the percentages of Iron (respectively between 42% and 69% and between 72% and 83%). The same trend can be observed for Zinc even if the difference between the two periods is less

**Table 5.** Average concentrations (± standard deviation) of the main chemical species for the three sampling sites and their relative variation between the period before lockdown (BL) and during lockdown (DL).

| Species | Period | FI-3 Average (ng m$^{-3}$) | FI-3 Var. % | FI-1 Average (ng m$^{-3}$) | FI-1 Var. % | PO-2 Average (ng m$^{-3}$) | PO-2 Var. % |
|---|---|---|---|---|---|---|---|
| PM | BL | $22500 \pm 4600$ | -25 | $23400 \pm 6500$ | -26 | $27900 \pm 6400$ | -28 |
| | DL | $17000 \pm 5000$ | | $17300 \pm 5200$ | | $20100 \pm 6100$ | |
| EC | BL | $1900 \pm 660$ | -66 | $820 \pm 310$ | -43 | $760 \pm 210$ | -37 |
| | DL | $650 \pm 220$ | | $470 \pm 120$ | | $480 \pm 160$ | |
| OC | BL | $6900 \pm 1800$ | -24 | $6700 \pm 3400$ | -30 | $8700 \pm 3300$ | -29 |
| | DL | $5300 \pm 2000$ | | $4700 \pm 1800$ | | $6100 \pm 2000$ | |
| S | BL | $390 \pm 220$ | +50 | $480 \pm 220$ | +36 | $460 \pm 190$ | +43 |
| | DL | $580 \pm 260$ | | $650 \pm 240$ | | $660 \pm 310$ | |
| Ca | BL | $2420 \pm 500$ | -38 | $1210 \pm 570$ | -11 | $1560 \pm 160$ | -12 |
| | DL | $1500 \pm 220$ | | $1090 \pm 110$ | | $1370 \pm 170$ | |
| Cr | BL | $19.3 \pm 6.8$ | -69 | $8.6 \pm 2.9$ | -68 | $7.4 \pm 2.0$ | -42 |
| | DL | $5.9 \pm 1.8$ | | $2.74 \pm 0.82$ | | $4.3 \pm 1.1$ | |
| Fe | BL | $1570 \pm 370$ | -66 | $660 \pm 270$ | -81 | $690 \pm 200$ | -52 |
| | DL | $530 \pm 160$ | | $128 \pm 59$ | | $330 \pm 110$ | |
| Cu | BL | $65 \pm 20$ | -74 | $29 \pm 15$ | -83 | $21.7 \pm 7.8$ | -72 |
| | DL | $16.5 \pm 6.0$ | | $5.0 \pm 13.3$ | | $6.0 \pm 2.8$ | |
| Zn | BL | $644 \pm 13$ | -40 | $27 \pm 10$ | -18 | $26.9 \pm 8.6$ | -24 |
| | DL | $26.3 \pm 6.6$ | | $21.8 \pm 8.2$ | | $20.4 \pm 5.3$ | |
| NH$_4^+$ | BL | $200 \pm 190$ | +206 | $230 \pm 220$ | +186 | $520 \pm 400$ | -2 |
| | DL | $640 \pm 240$ | | $670 \pm 240$ | | $510 \pm 200$ | |
| K$^+$ | BL | $147 \pm 96$ | +15 | $190 \pm 110$ | -8 | $322 \pm 98$ | -39 |
| | DL | $170 \pm 100$ | | $180 \pm 100$ | | $200 \pm 110$ | |
| NO$_3^-$ | BL | $1910 \pm 120$ | -26 | $1880 \pm 870$ | -53 | $4100 \pm 1700$ | -60 |
| | DL | $1430 \pm 690$ | | $890 \pm 550$ | | $1630 \pm 620$ | |

marked (between 18% and 40%). The decreases observed for Fe, Cu and Zn are in line with the interruption of vehicular traffic during the lockdown period. From these elements, markers of brake and tire abrasion (Gietl et al., 2010; Wik and Dave, 2009), a significant reduction in the source of non-exhaust traffic can be confirmed for all sites. The percentages of decrease of Fe and Cu are very high in the agglomeration of Florence (always above 66%) and slightly lower in the Prato site. As for the absolute values of the concentrations recorded in the two periods, it is important to note that in the urban traffic site (FI-3) these drop

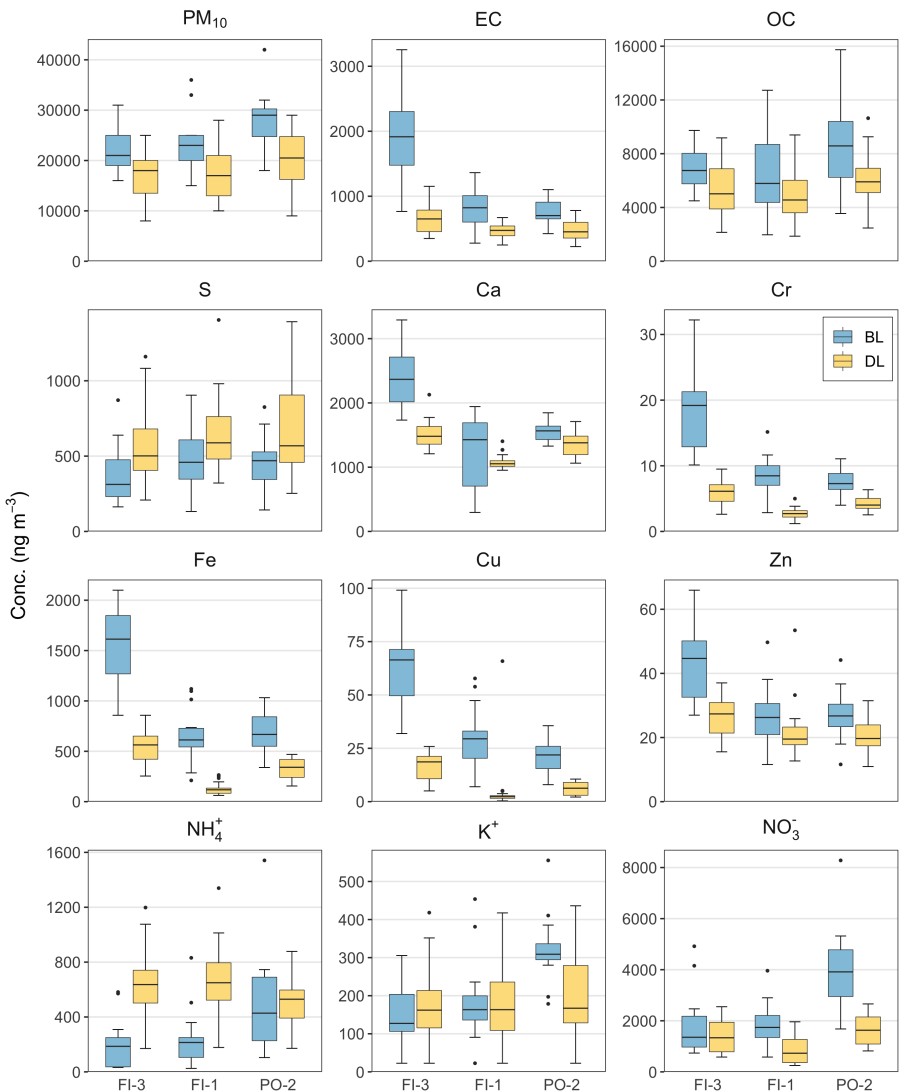

**Figure 7.** Comparison between the period before the lockdown (BL) and during the lockdown (DL) for the main chemical markers for each of the three sites. Boxes represent the interquartile range (IQR, between the 25$^{th}$ and 75$^{th}$ percentile) and the median (line inside the boxes), whereas whiskers cover the range of the samples over and under 1.5∗IQR and points are outliers.

significantly but during the lockdown they are slightly lower than those of the two urban background sites in conditions of unrestricted traffic that is, turning the traffic site into an urban background one.

For Calcium the difference between the traffic site and the other two background sites is even more evident. In all the sites there is a decrease in the averages between the two periods but, although for the background sites these decreases are very low (between -11% and -12%) and even lower than those of the total PM, for the traffic site it is greater (-38%). The relatively large

extent of such a decrease suggests that where the vehicular traffic is one of the major sources, the period of traffic restrictions has heavily affected the concentration of this element which is both a marker of natural mineral inputs and urban dust due to the resuspension caused by traffic or people strolling by. Also in this case, the absolute concentrations during the lockdown period on the traffic site drop to the values recorded in February in the two background sites.

Potassium and nitrate show a notable decrease in PO-2 (respectively -39% and -60%) and, only nitrate, also in FI-1 (-53%).

However, both species do not appear to be affected by the considerable reduction of the other species seen for the traffic site. In this case, ammonium seems to be influenced by the geographical position given the notable increase only in the two sites in Florence (with an increase of about three times) while in Prato it remains in the same range of values. The increase in ammonium concentration can be ascribed to agricultural activities that were not interrupted even during the national lockdown. Furthermore, this period was characterized by a condition of regional atmospheric stability with weak winds and an almost

complete absence of precipitation.

Furthermore, the increase in solar radiation also seems to have favored the accumulation of some pollutants of secondary origin. In fact, sulfur, present in the atmosphere mainly as sulfate, shows a substantial increase in all the sites (between 36% and 50%) with no particular differences between one site and the other.

The results obtained here with regard to the chemical composition are in line with that seen in the work of Massimi et al.

(2022) in which the impact of the lockdown in the city of Rome was assessed, taking into consideration three different sites in terms of position and type; taking into account the variability between the sites, the same variations were found between the periods before and during the lockdown, especially for elements coming from anthropogenic sources.

### 3.4  Sources identification and their changes

The PMF analysis was performed on a single dataset that combines the chemical composition data of the three sites described

in the previous paragraph (FI-1, FI-3, and PO-2), assuming that the profile of the main aerosol sources found by the model is the same in all the sites (Regione Toscana, 2021). In total 107 observations were used, and, according to the signal-to-noise ratio suggested by Polissar et al. (1998), 19 chemical species were selected in addition to the $PM_{10}$ mass: 13 species obtained from PIXE analysis, 3 from ion chromatography, organic and elemental carbon, whereas insoluble potassium was obtained from the difference between total potassium (from PIXE, not used in PMF) and soluble potassium (from IC). All the species

were selected as strong variables in the model and $PM_{10}$ was used as the total variable (with 400% of uncertainty).

PMF led to the identification of 6 sources, namely: secondary nitrate, combustion, traffic, marine, soil dust, and sulfate. These final factors were chosen after examining multiple solutions with a different number of factors (from 5 to 8) based on the real meaning of their chemical profiles, their temporal trends, and the residuals of all the chemical species. The ratio between Q and its expected theory value was 1.11. The correlation between the mass of the particulate matter measured by weight and

the mass reconstructed by the model was good (slope = 0.94 and $R^2$ = 0.98). The study of uncertainties was also performed using the bootstrap (BS) and displacement (DISP) method. For BS the number of factors correctly mapped for 100 runs (with a correlation threshold equal to 0.6) was never under 99%, whereas for DISP no swaps were observed between the factors and the largest decrease in Q was -0.031.

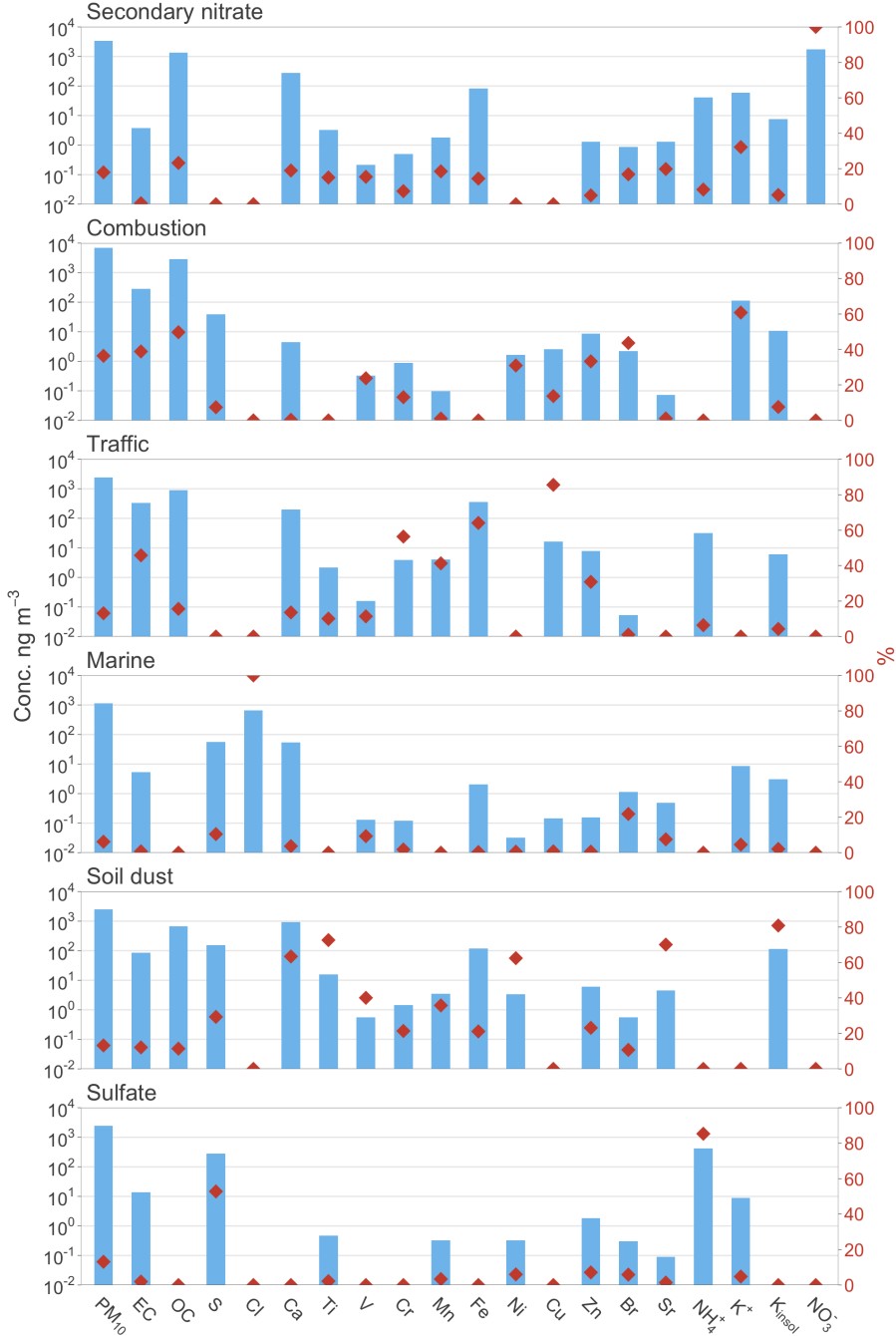

**Figure 8.** Chemical profiles of the factors obtained by PMF analysis, showing the mass contribution in $\mathrm{ng\,m^{-3}}$ of each species within each source (blue bars) and the average percentage contribution of each source to the total concentration of each element (red points)

The chemical profiles of the sources found in the final solution are shown in Fig. 8, which shows the average concentrations of the elemental contributions in each factor (blue bars, left axis) and their percentages relative to the total concentration of the element (red dots, right axis).

The assignment of the physical source corresponding to each factor was done basing on the markers of their chemical profile. The first factor is called secondary nitrate because it is mainly composed of $NO_3^-$ and $NH_4^+$. It is associated with a secondary component produced by local combustion processes, mainly vehicular emissions but also domestic heating (Nava et al., 2020), which are the main sources of $NO_x$, in turn oxygenated to $NO_3^-$ (Seinfeld, 1986). The presence of calcium in the chemical profile of this source is due to the formation of the salt $(CaNO_3)_2$ which forms in dry conditions and in the presence of $CaCO_3$ (Amato et al., 2016). Another important component is OC, representing the condensation of semi-volatile organics on the high specific surface area of the ammonium nitrate particles.

In the second factor the high concentrations and percentages of elemental, organic carbon, and soluble potassium (respectively 39%, 50% and 61%) allow to recognize this source as biomass burning for domestic heating (Nava et al., 2015; Sharma et al., 2016; Piazzalunga et al., 2011). The OC/EC ratio (about 11) in its profile is within the ranges reported in literature for a biomass burning source (Vincente and Alves, 2018). The metropolitan area of Florence and Prato is well supplied with natural gas, but in the suburbs on the hills and also inside the residential areas there are many chimneys and recently, due to tax incentives, many citizen houses are equipped with small wood or pellet stoves (Nava et al., 2020). In quantitative terms, this is the source that gives the highest contribution to the mass of the particulate.

Factor three has been associated with urban traffic, since it is mainly composed of OC and EC and traced by specific elements like Fe, Cu and Zn as reported in the literature (Viana et al., 2008; Ntziachristos et al., 2007; Matthaios et al., 2022). It includes both exhaust and non-exhaust emissions. Cu and Fe in this factor (86% and 64% respectively), Ca, Mn, and Zn, are associated with vehicle's wheel and brakes wear, and the resuspension of urban dust particles due to vehicular traffic (Harrison et al., 2012; Thorpe and Harrison, 2008; Charron et al., 2019; Handler et al., 2008; Gillies et al., 2001). The OC/EC ratio (about 3) is similar the one found in other urban background sites (Amato et al., 2016).

Although it was not possible to analyze Na, in the fourth factor the high loading of Cl, which is totally assigned to this factor, allows to recognize the marine source. The analysis of the back-trajectories allowed to confirm the attribution of this source since the sea spray comes from the Tyrrhenian Sea carried by strong west winds, while the arrival of marine particles from the east is forbidden due to the presence of the Apennine mountains. S and Ca in this factor are due to their presence in the sea salt (Bertram et al., 2018) and were already found in other works (Manousakas et al., 2021; Gugamsetty et al., 2012).

The fifth source is identified as soil dust, since it includes all the crustal elements that have been analyzed: Ca, Ti, Mn, Fe, Sr, and insoluble K. These elements are the major constituents of airborne soil and road dust; Ca may be also due to local construction activities.

Finally, the sixth factor contains almost exclusively S and ammonium, with these two species showing also a high explained percentage in this factor (respectively 53% and 85%). Although sulfate ion concentrations were available from ion chromatography (which has a good correlation with elemental sulfur) total sulfur was used for PMF analysis because it gives information also on sulfur which is not in the form of sulfates. This factor was then referred to as sulfate since it was seen that this chemical

**Table 6.** Average concentrations ($\pm$ standard deviation) of the six sources for the three sampling sites and their relative variation between the period before the lockdown (BL) and during the lockdown (DL).

| Source | Period | FI-3 Average (ng m$^{-3}$) | FI-3 Var. % | FI-1 Average (ng m$^{-3}$) | FI-1 Var. % | PO-2 Average (ng m$^{-3}$) | PO-2 Var. % |
|---|---|---|---|---|---|---|---|
| Secondary nitrate | BL | $2200 \pm 1400$ | -24 | $2200 \pm 1000$ | -56 | $5000 \pm 1900$ | -59 |
| | DL | $1700 \pm 830$ | | $960 \pm 690$ | | $2030 \pm 890$ | |
| Combustion | BL | $5400 \pm 3800$ | +58 | $8800 \pm 4600$ | +15 | $10500 \pm 3900$ | -18 |
| | DL | $8600 \pm 3500$ | | $10100 \pm 3700$ | | $8600 \pm 2600$ | |
| Traffic | BL | $9500 \pm 2500$ | -77 | $3700 \pm 1900$ | -99 | $3000 \pm 1100$ | -80 |
| | DL | $2190 \pm 920$ | | $50 \pm 120$ | | $600 \pm 430$ | |
| Marine | BL | $2400 \pm 2700$ | -88 | $3300 \pm 6000$ | -98 | $2700 \pm 4900$ | -93 |
| | DL | $280 \pm 310$ | | $70 \pm 100$ | | $190 \pm 270$ | |
| Soil dust | BL | $2310 \pm 670$ | -1.7 | $1110 \pm 940$ | +73 | $860 \pm 370$ | +165 |
| | DL | $2270 \pm 360$ | | $1919 \pm 260$ | | $2270 \pm 420$ | |
| Sulfate | BL | $560 \pm 850$ | +437 | $960 \pm 980$ | +249 | $2200 \pm 1400$ | +20 |
| | DL | $3000 \pm 1200$ | | $3300 \pm 1200$ | | $2600 \pm 1100$ | |

species is closely associated with ammonium as in the atmosphere they neutralize each other by forming salts, and furthermore, sulfate is a species always present as background in this area as a secondary product of the oxidation of $SO_2$ (Gen et al., 2019). In other past works it has been seen that in Tuscany it is a regional background, with similar values throughout the regional territory (Regione Toscana, 2021).

As for the comparison made for the chemical species in the previous paragraph, even for the sources the samples were divided between before and during the lockdown. Table 6 shows the averages of the contribution of the sources in the two periods and their relative variation for each of the three sites, while Fig. 9 shows the distribution of values and in Fig. 10 both the concentrations and the percentages of the sources in the two periods are compared.

The two factors that show a significant variation in all monitored sites are those related to marine and traffic sources. Regarding the marine source, since it is natural, we can exclude that its strong decrease between the two periods is correlated with the lockdown restrictions. From the calculated back-trajectories, it has been seen that in this case such a big difference could be due to the change in the atmospheric transport. Before the lockdown, the air masses that reach the sampling sites mainly came from the Tyrrhenian Sea and remain at low altitude while, during the lockdown, they came from the opposite direction reaching the sampling sites from higher altitudes, thus bringing air masses with lower content of marine aerosol. The difference observed for the traffic source is instead a direct consequence of the restrictions on vehicular circulation imposed by the government to reduce the spread of the virus. For the urban background sites FI-1 and PO-2 the average reduction of this source is respectively 99% and 80% while FI-3, which is an urban traffic site, exhibited a decrease of 77% and recorded

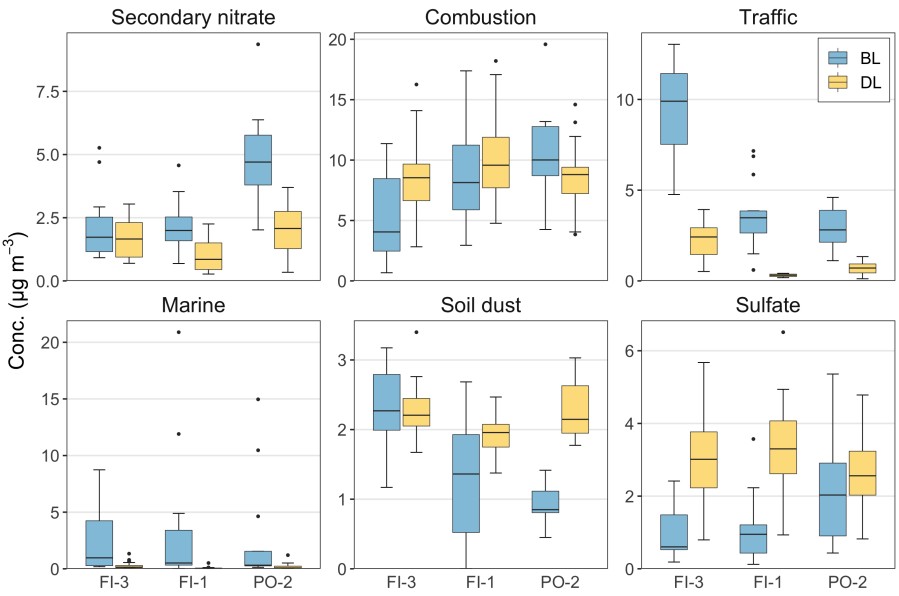

**Figure 9.** Comparison between the period before the lockdown (BL) and during the lockdown (DL) for each source obtained by PMF analysis for each of the three sites. Boxes represent the interquartile range (IQR, between the 25th and 75th percentile) and the median (line inside the boxes), whereas whiskers cover the range of the samples over and under $1.5*$IQR and points are outliers.

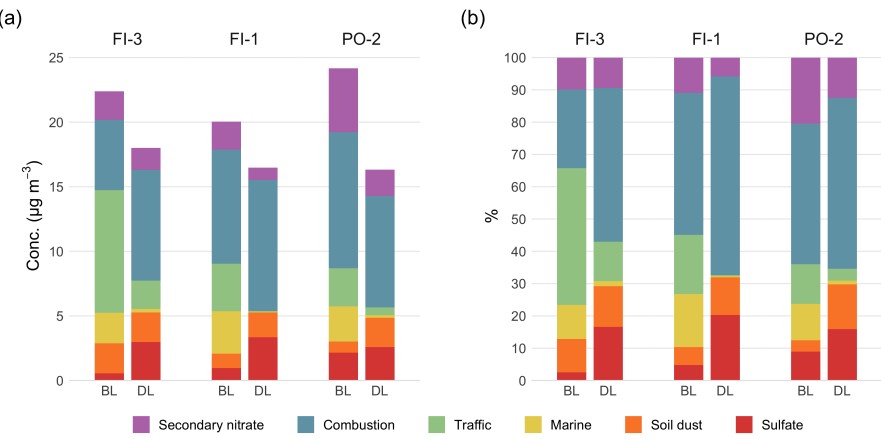

**Figure 10.** Comparison between the two periods for the concentrations and percentage of the sources obtained by PMF analysis for each of the three sites.

the greatest decrease in absolute concentration. In this latter site, the weight of the traffic source on the total particulate load has considerably reduced between the two periods, lowering his contribution from 42% of $PM_{10}$ (the main source) to only

12% (Fig. 10b). It is possible that such a strong reduction in the traffic source in FI-1 which is mainly residential is due to the presence of some schools in that area that were closed during the lockdown period.

The results obtained for the traffic source were then compared with the data provided by the National Autonomous Company of State Roads (ANAS) and by Google. ANAS, thanks to its automatic statistical traffic detection system, in the March report of its Traffic Observatory reports a 53% reduction in total vehicle traffic on the main state roads in Tuscany compared to the previous month, while in the April report the monthly reduction is 43%, leading to a total reduction of 73% in April compared to February (ANAS, 2021). The data presented by Google, on the other hand, are related to the mobility of users anonymously

detected for the services normally offered by the platform and made public during the pandemic period for study purposes (Google, 2021). For the comparison with the results of this work, the data corresponding to the same days of the analyzed samples was used considering the mobility that was classified towards workplaces and public places such as public transport hubs (i.e., subway, bus, and train stations), recreational and retail businesses (i.e., restaurants, cafes, shopping centers), and public parks, but excluding travels to supermarkets, grocery stores and pharmacies. Taking as the starting baseline the average

of the days from 3 January to 6 February 2020 (baseline decided by Google), these data show that in Tuscany travels have reduced on average by 76% between the lockdown period and the previous one. Although the data used in these analyses present some internal approximations since from ANAS come counts made only on major roads and from Google the tracking of connected phones to and from places of interest, they proved to be consistent with the results for the traffic source identified through PMF which has an average decrease of 85%.

The nitrate source is also observed to decrease for all sites, probably affected by rising temperatures, but with not significantly high variations. The comparison of the trends of this secondary source and $NO_2$, its main primary precursor, shows a different behavior for each of the three sites: PO-2 with -59% shows the highest decrease in secondary nitrate, which is more evident as it started from already very high concentrations, but only -37% for $NO_2$; in FI-1 the two values are very close (respectively -56% and -50%); instead, at FI-3 the secondary sources have a variation of only -24% but $NO_2$ decreases by 39%. Therefore, if on the

one hand $NO_2$ decreases similarly in the three sites, this does not happen also for its secondary products that collapse in the two underlying sites but do not have a significant decrease in the traffic site, which therefore shows inertia in the removal of these secondary products. The pattern is probably due to the complexity and non-linearity of the processes that in the atmosphere lead to the formation of secondary nitrate starting from primary $NO_x$.

  The combustion shows an evident increase only at the FI-3 site but the distribution of concentration values does not show a

particular difference between the two periods at any of the sites. During the lockdown period, combustion becomes the main source of particulate matter for the FI-3 site (48%) and for the others it intensifies its influence, reaching 62% for FI-1 and 53% for PO-2 (Fig. 10b).

  The pattern observed for secondary nitrates is explained by observing its behavior together with those of the traffic and combustion sources, which are the two sources that have a greater weight in the formation of $NO_x$, the precursor of nitrates in

the atmosphere. In the PO-2 site a decrease in both the traffic and the combustion sources is observed and, together with these two strong reductions in emissions, also the secondary nitrates are affected with a significant decrease in their atmospheric concentration. On the other hand, as regards the FI-3 site, the combustion source, which increases during the lockdown, is in

contrast with the trend of the traffic source, which is significantly decreasing; this opposite trend of traffic and combustion results in a maintenance of the same concentrations of secondary nitrates. The third site (FI-1) shows an intermediate behavior:

in fact, with a very strong reduction of the traffic source and a modest increase in the one related to combustion, it shows a marked reduction of secondary nitrates. Although the formation of secondary nitrates in the atmosphere is mainly triggered by the photochemical activity of solar radiation, which in the period considered for this work was particularly favorable for this type of reactions especially since it is combined with atmospheric stability, this behavior confirms the direct connection of each of the two primary sources with the secondary nitrate formation (Fan et al., 2020; Zhao et al., 2020b).

The factor representing the crustal fraction of the particulate matter is the one showing the greatest difference between the two periods in the three sites. As it can also be seen from Fig. 9, it spans from a fairly constant contribution at FI-3 to an increase of 165% at PO-2. Being a natural source, this inconsistent behavior among the three sites may be ascribed to variability in the local sources of crustal dust around the sampling sites. Thanks also to the conditions of atmospheric stability in the region, the only evident effect of the restrictions on this source was the leveling of its contribution, bringing it to almost constant values for

all the sites and canceling those initial differences that could be due to different intensities of resuspension caused by different levels of traffic and anthropogenic activities.

Finally, the sulfate source, which consists almost exclusively of sulfur and ammonium, is the only one that shows an increase in all sites, even if almost irrelevant in the PO-2 site. On the other hand, the increase in the two Florence sites was considerable, reaching an average of +440% and +250%, which is an increase also favored by the rising of temperatures and solar radiation,

which facilitates gas-to-particle conversion.

### 3.5 Comparison with past campaigns

Two of the three sites for which the chemical composition of the particulate matter was analyzed for this work were used in the past in two sampling campaigns. FI-3 is one of the sites used in 2006 in the PATOS regional project (Nava et al., 2015), while FI-1 was used in 2013 in the AIRUSE European project (Amato et al., 2016). By using the data of these two campaigns,

it was therefore possible to perform a comparison with the samples of this work collected in 2020. The comparison (shown in Fig. 11) was made only in the days of the year covered by the samples of 2020 (i.e., from February to April) which were divided, as described above, after 10 March of each considered year. For both campaigns the source apportionment with PMF had been performed. The chemical profiles of the sources of this work are consistent with the analogous sources of the past two campaigns, although different numbers of factors were found by the PMF analysis. The shown comparison refers only to

the common sources found in the campaigns.

Leaving aside the comparison of natural sources which are almost exclusively subject to variations due to atmospheric transport conditions such as the wind intensity and the origin of the air masses, as regards the anthropogenic sources, the greatest difference is observed for the traffic. The large decrease described above is in sharp contrast to the almost complete absence of change in concentrations in the previous years, where, although the median value changes, the distribution of values

remains approximately within the same range. As regards the combustion source, the slight increase observed in both FI-1 and FI-3 is in contrast with the trend of the 2013 campaign in which a decrease has been observed in the same period. The

cause is probably due to the increased use of domestic heating by citizens forced home by the restrictions. Finally, secondary sulfates and nitrates show a clear change between the two periods of 2020 but their behavior does not differ significantly with what was observed in the previous campaigns. The comparison with these two past campaigns confirms and strengthens the interpretation that the observed effects, especially for primary anthropogenic contributions, are related to the lockdown and not to the weather conditions of March and April 2020.

## 4    Conclusions

To control the rapid spread of the SARS-CoV-2 virus, in March 2020 Italy imposed national lockdown policies. With the social distancing imposed for health reasons, many urban and industrial activities had drastically decreased, even to a halt. Among these, the collapse of vehicular traffic was one of the most important consequences observed in the weeks following the lockdown. So, one of the side effects of all these restrictions has been the improvement of air quality, especially in urban centers. These particular conditions offered a unique and powerful window of opportunity to study air quality in the absence of a major source of air pollution in urban areas.

This work evaluated the impact of the conditions of almost no traffic and other anthropogenic sources on the air quality in Tuscany between March and April 2020, in particular in the metropolitan area of Florence thanks to the chemical characterization of particulate matter and the study of its sources obtained implementing the PMF receptor modeling. The variations of the particulate matter load of its chemical composition and of the source contributions were then investigated by comparing the weeks of lockdown with those immediately preceding and, where possible, with previous years.

The comparison between the periods before and after the lockdown shows that the concentrations of Fe, Cr, Cu, and, albeit milder, also of Ca have been significantly reduced in all the sites.

Since these elements are linked to the consumption of brakes and tires and the resuspension of urban dust due to vehicular traffic, it follows that even in the distribution of sources, the factor recognized as traffic shows a decrease that has been quantified between 77% and 99%. In the same period, we also see a sharp decrease in $NO_2$ compared to the previous three years. However, the strong reduction in traffic and its related pollutants (i.e., marker elements and $NO_2$) did not lead to a drop in the total $PM_{2.5}$ load. Even the concentration of $PM_{10}$, although decreased in the months of lockdown, had not undergone major changes that can be attributed to the decrease in traffic: in the case of urban background sites because the traffic does not represent a large fraction of the total, while in the traffic site because its decrease was compensated by the increase in sulfates and combustion particulates (biomass burning). The sulfate source is in fact the $PM_{10}$ fraction which has increased for all sites thanks to the increase of solar radiation and to the period of atmospheric stability in the months of March and April.

It can be hypothesized that the relatively minor decrease in particulate matter compared to traffic pollution is due both to the presence of large concentrations of precursor pollutants, such as ammonia from agriculture, in a concentration sufficient to produce PM of secondary origin together with sulfate, and high consumption for domestic heating, in meteorological conditions that limited the dispersion of pollutants and produced emissions of the primary component.

Despite the evident reduction in traffic pollution, the presence of other sources, and in particular the presence of particulate matter of secondary origin, which has different formation and removal rates from those of PM of primary origin, makes it difficult to evaluate the effects of the lockdown on the presence of pollutants in conditions of almost total reduction of some of the anthropic activities typical of urban areas. Also for this reason, this study confirms the complex nature of atmospheric pollution even in the case of isolation of an important urban source of primary emission. The exceptional nature of the event certainly helps to isolate some sources of particulate matter but some dynamics still remain to be understood, especially about the removal of particulate matter of secondary origin. However, it should be noted that the period elapsed since the implementation of the restrictions to cope with the spread of the SARS-CoV-2 virus is restricted to a few months of a single season and they cannot give indications on the effect that such restrictions could have had in another period of the year when the loads of the sources could be different.

*Author contributions.* FL and BPA planned the campaign; SN, GC, GP, MC, AA, FG, AF, CC, EF, and GN performed the measurements; FG, AF, CC, EF, and GN analyzed the data; FG, AF, CC, EF, and GN wrote the manuscript draft; FL, BPA, EF, RT, MS, and SB reviewed and edited the manuscript.

*Competing interests.* The authors declare that they have no conflict of interest.

*Acknowledgements.* We thank Dennis Dalle Mura, Roberto Fruzzetti, Tiziana Cecconi, Marco Stefanelli and Stefano Fortunato (ARPAT) for PM$_{10}$ and PM$_{2.5}$ sample collection and for managing the AMS described in this study.

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

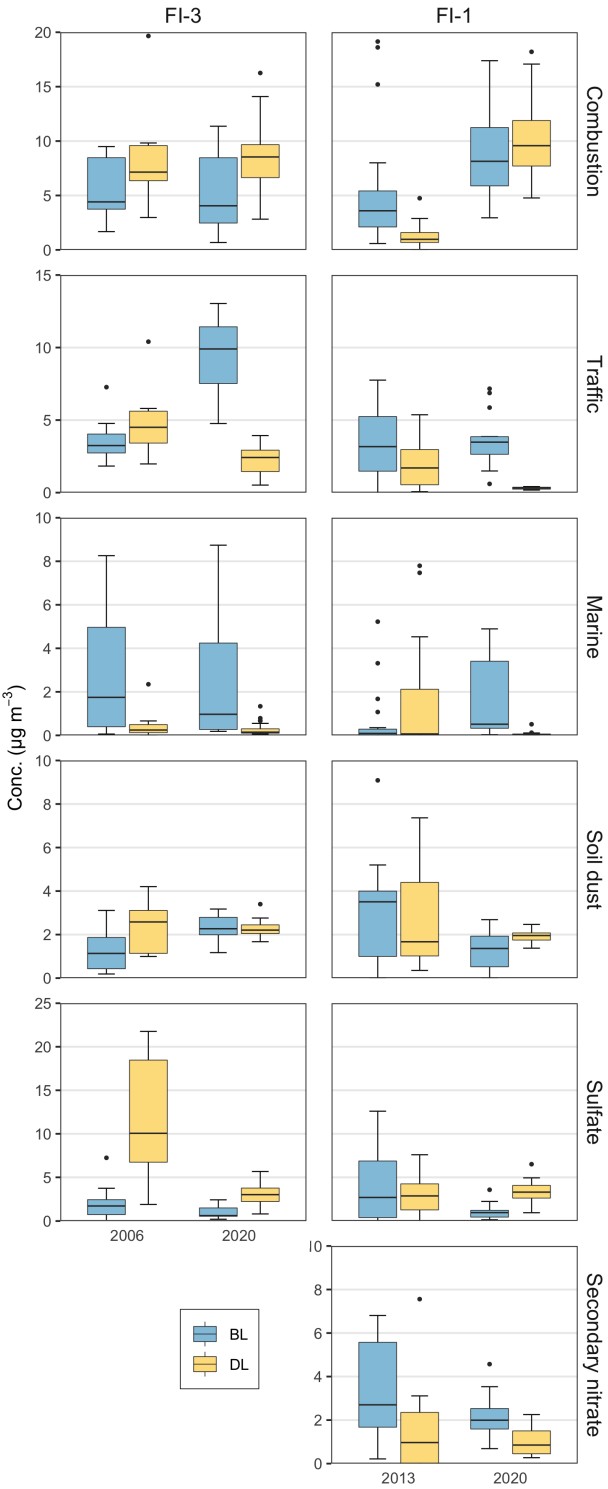

**Figure 11.** Comparison between the period before the lockdown (BL) and during the lockdown (DL) for the 2020 samples and those collected in two past campaigns: the PATOS regional campaign in 2006 for the FI-3 site and the AIRUSE European campaign in 2013 for the FI-1 site.