# Peer review of "PM10 variation, composition, and source analysis in Tuscany (Italy) following the COVID-19 lockdown restrictions"

_Atmospheric Chemistry and Physics, 2021_

## Author Response (AR1)

Response to referee comment 1:

Regarding the manuscript entitled "PM10 variation, composition, and source analysis in Tuscany (Italy) following the COVID-19 lockdown restrictions". The manuscript presents new results for the effects of the lockdown period in the particulate-related (mainly) pollution of a region in Italy. Generally, the manuscript is well structured, interesting, and should be considered for publication, but it is in need of major revisions. Some of the decisions regarding the source apportionment analysis need to be discussed more in the manuscript. By saying that, I do not imply that I disagree with the decisions the authors made, but I strongly believe that the readers would benefit if such a discussion were included. Additionally, I believe that sections 3.5 and especially 3.4 need to be thoroughly revised. Section 3.4 is the heart of the manuscript, and it appears extremely rushed, with little to no discussion and just a couple of references to support it. This section needs to be re-written and the results should be compared with that of other covid lockdown period publications. I would also suggest increasing the discussion and comparison to other studies in section 3.3, but that is just a suggestion. I really appreciate the amount of work that was put into the study, regarding the different types of analysis that were done and the data collection, and I feel that it is a pity not to showcase that in the discussion of the results.

> **Sections 3.3, 3.4 and 3.5 have been modified and expanded with more in-depth discussions of the results, adding also other references and comparisons with other works.**

Line 86: Why was the pre-lockdown period preferred as the standard for the comparison compared to, for example, the same time period of the previous year? Could the results be affected by the seasonal effect on source emissions (winter to spring)?

> **Unfortunately, only samples from the same year were available as PM is not routinely collected for chemical analysis every year, but only mass is continuously measured by beta attenuation. The comparison of the masses of 2020 with the previous years is in fact reported in the text. We are aware of possible seasonal effect and to separate the seasonal effects we made the comparison with previous campaigns.**

Line 122: Why was this protocol used instead of EUSAAR2? Is there a certain advantage?

> **The NIOSH protocol has been found to be more reliable for EC quantification because in urban areas, protocols reaching high temperatures in He atmosphere are preferable to low-temperature ones to get rid of typical interferences which affect EC results (Piazzalunga et al. Atmos. Chem. Phys., 11, 10193–10203, 2011)**

Lines 134-135: When samples from several sites are pooled together, the assumption that is made is that the number and type of sources is the same at all sites. Even though this is a valid approach that has been proven effective on many occasions in the past, in my opinion, a short discussion of why this is a reasonable assumption for the sites of the study would be very useful to the readers. For example, another approach could be (even though the limited number of samples here makes it hard to implement) to separate the sites into traffic and urban background and do two different SA analyses since traffic-related source profiles might be different in the two regions. Another option could be to separate the samples as pre and during lockdown and examine the differences in the profiles as well as the contributions. Generally, I am not suggesting that the option used was not good, but I believe it would be good to have a discussion here why it was preferred since there are other options that appear valid and may offer some advantages.

**We think that the use of a subset of samples would lead to disadvantages in the validity of the result of the source apportionment because of the already limited number of samples available for PMF analysis; PMF works better when there is a high variability in the source contributions between samples. Less than 50 samples is not a sufficient number of samples to obtain a reliable PMF analysis. Furthermore, it has been seen that the metropolitan area of Prato and Florence is homogeneous in terms of type and profile of sources. Prato and Florence are considered a single extended metropolitan area, as one passes from Florence to Prato without interruption. We have modified the sentence and added a reference in the article.**

Lines 136-138: Has the rotational ambiguity and the stability of the solution been evaluated?
**The stability of the solution was tested with bootstrap and displacement. Two sentences have been added in the text to explain it.**

Line 268: How much uncertainty was assigned to PM?
**The uncertainty used is 400% which is recommended for the total variable. This information has been added in the text.**

Line 274: Swaps are observed for displacement only. For BS it would be useful to report the number of BS runs and how many of them correlated to the base case and what was the correlation threshold used. Which species have been actively displaced during DIS ("good" species)? Generally please include the mathematical indicators that prove the goodness of the fit.
**Thanks for the suggestion. The sentence was changed in the text to differentiate between bootstrap and displacement and to insert the missing information. For BS, 100 runs were done and the correlation threshold used is 0.6. All good species were used for displacement. We had already reported the correlation between the PM10 measured mass and the mass reconstructed by the model (slope = 0.94 and R2 = 0.98).**

Line 280: Please rephrase. Secondary species are formed in the atmosphere, their precursors can originate from a source, not them directly.
**Thanks for the comment. The sentence has been rewritten.**

Lines 280-282: This statement is unclear. Please explain/rephrase
**The sentence has been rewritten to make it more understandable.**

Lines 287-290: No references are provided here.
**References added: Harrison et al. (2012), Thorpe and Harrison (2008), Charron et al. (2019), Handler et al. (2008), Gillies et al. (2001).**

Lines 290-292: Could you please explain the presence of Ca and S to the factor? Generally, the discussion here appears rushed, with little to no discussion and no references. The use of the English language is also not on par with the previous sections.
**The sentence has been rewritten. Ca and S were already found in this source in other works (references have been added to the text). We have modified all the part regarding the identification of the sources, with a deeper discussion and with the insertion of many references.**

Lines 295-296: Could you please justify that decision? In theory, sulfate is preferable to sulfur since it consists a higher percentage of PM mass.

**We used elemental sulfur because S could not only be in the form of sulphate, therefore, in principle, elemental sulfur gives more information. This explanation has been added to the text.**

Line 311: urban or urban background? 99% seems an extremely drastic decrease even for a lockdown! Can this be an effect/artifact of the analysis and particularly due to the fact that the samples from all sites were combined? Providing the time series of the source contributions would help to assess how good the solution is.

**They are both urban backgrounds (this has been corrected in the text). These are residential areas where there are also some schools which produce the peaks in traffic intensity; we think that the result is plausible, given all the schools were closed and we observed an almost total absence of traffic during the lockdown in FI-1.**

Lines 312-314: Please rephrase

**The sentence has been rewritten.**

Lines 324-327: A 99% decrease is not consistent with a 75% decrease in vehicular volume. As mentioned before the 99% is hard to support and appears to be an artifact of the analysis.

**The decrease in traffic provided by Google in the whole metropolitan area of Prato and Florence (75.6%) is an average value; it is close to the average reduction of the traffic source in three sites considered (85.3%). We consider it a good result given how far these data are from each other (geolocation of phones vs receptor model). The effect at FI-1 can be justified as mentioned above by the type of site (residential with the presence of schools) with practically no traffic during the lockdown period (it has been added to the text).**

Lines 328-329: This is the reason that comparing the same time period of 2019 to that of the lockdown period of 2020 might have worked better. In this case, you need to take into account the seasonal effects as well.

**Unfortunately, it was not possible to compare the same period of the previous year because the samples were not available. The sentence has been corrected.**

Lines 335-336: This explanation is not enough. Please try to provide a better explanation for this, or/and compare it to other studies from the same time period in different regions. In the entire section 3.4, which is by far the most important of the entire study there are only 3 references.

**A paragraph has been added to further discuss the behavior of secondary nitrates**

Lines 343-344: What does this mean exactly? What could the local natural sources of soil dust be, and why are they different in areas with such close proximity? The source of dust is practically one, resuspension (other such as volcanoes are possible, but it is not relevant to the area). Resuspension can be other natural (wind-related) or can also be induced by vehicular movement. As said before, the discussion in this section appears very rushed, and the explanations provided for the observations are vague.

**This paragraph has also been expanded with a more in-depth discussion.**

Lines 357-365: How do the source profiles compare? Any differences there? Comparing the source profiles can be much more interesting than comparing the source contributions.

**The profiles of the sources obtained in this study are consistent with those obtained in the other two campaigns with which they are compared. We think that the comparison of the source contributions strengthens the interpretation that the observed effects, especially for primary anthropogenic contributions, are related to the lockdown and not to the weather conditions of March and April 2020.**

Response to referee comment 2:

Well written paper on comparisons between pre during and post COVID time frames at a range of sampling sites in Italy. Comprehensively covers the sampling and the analysis aspects. The data are sufficiently novel to warrant publication.
The general comments that I have are mainly related to:
Better clarification and discussion around which samples were PM10 and which were PM2.5 in each section. I found this distinct often hard to unravel when certain sections were discussed and this is particularly important.

**$PM_{2.5}$ is only discussed in one paragraph. A sentence has been added in that paragraph to clarify this.**

I know PM10 contains PM2.5 but the discussion around the primary use of PM10 in the PMF analysis is weak and not convincing. Many of the sources obtained are PM2.5 focused, like vehicles and secondary sulphate so why use PM10? Needs more convincing discussion. Also key elements like Si and Al that drive soil which is a key component of PM10 were no measured. The effect of not having these elements included is not adequately discussed
PMF is a powerful source apportionment tool but it only ever gives average source fingerprints across the data supplied, small changes in these fingerprints can produce significant changes in their contributions to the total PM10 mass. What would the sources look like if only the PM2.5 masses were considered – after all the major changes occurred in the Traffic signal which should clearly be in the PM2.5 fraction?

**The mass of $PM_{10}$ and $PM_{2.5}$ were measured by beta attenuation with an automatic instrument, but the chemical analyzes were performed on dedicated samples, which were sampled only for $PM_{10}$.**
**We are aware that Si and Al are important markers of the soil source, however we measure other elements, such as Ca, Ti, Fe which are also important markers of the same source. Indeed, the PMF was anyway able to identify this source.**

Cl is often present in Traffic and Combustion fingerprints to associate it only with sea spray when Na was not measured needs more justification. Maybe 7 fingerprints not 6 in the PMF fit would tease this out!

**Several solutions have been tried, even with a higher number of factors. However, by adding a factor we obtained the separation of sulfur from ammonium in the secondary sulphate factor. Considering the presence of sulfur, Br and the analysis of the air masses back-trajectories that show that this source is associated to air masses coming in a short time from the Tyrrhenian Sea, we think it is reasonable to associate the marine source to this factor. We have modified the sentence.**

I didn't see any meaningful discussion and quantitative data on the Q values for the PMF fits wer they close to the expected theory values?

**The ratio between Q and its theoretical value is 1.11. It has been added in the text.**

All the scales on many of the plots and figures are too small to read easily should be increased in size.

**The figures have been corrected to make them more readable.**

The conclusion alludes to the fact the number of samples in this study covering just a couple of months might be too small to draw major conclusions! Do the authors believe this?

**The meaning of this last sentence was not clear. Obviously, the exceptional nature of the event certainly helps to isolate some sources of particulate matter but some dynamics still remain to be understood, especially with regard to the reduction of particulate matter of secondary origin (as also seen in other published works). For this reason, the short period of lockdown restricted to a few months of a single season and in a period when the concentration of particulate matter is lowered by the change in the atmospheric conditions does not allow to reach more precise conclusions. The sentence has been changed to make it clearer.**

---

## Author Response (AR2)

All changes requested by the editor have been made.
In addition, we add some notes:

Lines 29 and 285: Replace "anthropic" by "anthropogenic".
"anthropic" was found only on line 29.

Lines 270 and 321, on one hand, and 574, on the other hand: The years are inconsistent.
Lines 343 and 459: The years are inconsistent.
In these two cases, the date in the reference was only that of the last access. Since the data contained in the web pages referred to different years, the correct year was added at the end of the reference.

---

## Author Response (AR3)

Your comments of 11/6/22:

Lines 8 and 217: Replace "thermo-optical" by "thermal-optical".
    *Corrected*

Line 9: Replace "PIXE" by "particle-induced X-ray emission" and replace "PMF" by "Positive Matrix Factorization".
    *Corrected*

Lines 29 and 285: Replace "anthropic" by "anthropogenic".
    *"anthropic" was found only on line 29.*

Lines 51 and 378: There are two "Zhao et al., 2020" in the list of References; they should be renamed there to "Zhao et al., 2020a" and "Zhao et al., 2020b" and appropriate alterations should be made in lines 51 and 378.
    *Corrected*

Line 65: Replace "VOCs emissions reduction" by "reduction of emissions of volatile organic compounds".
    *Corrected*

Line 71: Replace "analysis of" by "analyses of".
    *Corrected*

Line 73: Replace "that the variations" by "that variations".
    *Corrected*

Line 102: Replace "in ultrasonic" by "in a ultrasonic".
    *Corrected*

Line 107: Replace "PIXE" by "particle-induced X-ray emission (PIXE)".
    *Corrected*

Line 121: Replace "thermo-optical" by "thermal-optical transmittance".
    *Corrected*

Tables 5 and 6, for the Var. % data: most percentages have too many significant figures; two significant figures suffice and if the first significant figure is "1" then three significant figures can be used.
    *Corrected*

Line 265: Replace "was found" by "were found".
    *Corrected*

Lines 270 and 321, on one hand, and 574, on the other hand: The years are inconsistent.
    *The date in the reference was only that of the last access. Since the data contained in the web pages referred to different years, the correct year was added at the end of the reference.*

Line 284: Replace "elementary" by "elemental".
    *Corrected*

Line 287: Replace "associate" by "associated".
    *Corrected*

Lines 294, 302, 315, and 350: the percentages given here have too many significant figures; two significant figures suffice; thus, replace "38.9%, 49.7% and 60.8%" by "39%, 50% and 61%", and so on.
    *Corrected*

Line 298: Replace "citizens are" by "citizen houses are".
    *Corrected*

Line 301: Replace "in literature" by "in the literature".
    *Corrected*

Line 329: Replace "atmospheric currents" by "atmospheric transport".
    Corrected
Line 330: Replace "maintain a low" by "remain at low".
    Corrected
Lines 343 and 459: The years are inconsistent.
    The date in the reference was only that of the last access. Since the data contained in the web pages referred to different years, the correct year was added at the end of the reference.
Line 389: Replace "conversions" by "conversion".
    Corrected
Lines 404-406: What is written here disagrees with what can be seen in Fig. 11; for Combustion in FI-1 there is a decrease for 2013 and not for 2020. The text should be modified.
    Corrected
Line 407: Replace "do not differ" by "does not differ".
    Corrected
List of References: Replace ". doi" and ", doi" by ", http://doi" throughout.
    Corrected
Line 461: Replace "Spain). Sci." by "Spain), Sci.".
    Corrected
Line 499: Insert a space before "Cao".
    Corrected
Line 500: Replace ",125" by ", 125, e2019JD031998".
    Corrected
Line 515: Replace "February)" by "February), 115682".
    Corrected
Line 529: Replace "12(2)" by "12(2), 190".
    Corrected
Lines 578-578: This reference is incorrectly to a review of the book of Seinfeld; "Pollution, Environmental Science & Technology, 20(9), Wiley, New York, NY, USA. doi.org/10.1021/es00151a602, 1986." should be replaced by "Pollution, Wiley, New York, NY, USA, 1986."
    Corrected
Line 590: Replace "September)" by "September, 110193".
    Corrected
Lines 597-599: This reference should be moved up to before "Mor et al., 2021".
    Corrected
Line 624: Replace ", Zhang" by ", and Zhang".
    Corrected
Line 625: Replace ", 242" by ", 242, 117762".
    Corrected
Line 629: Replace ", 739" by ", 739, 140000".
    Corrected

Your comments of 24/6/22:

Line 442: Replace "anthropic" by "anthropogenic".
Corrected